# Monitoring HIV-1 Assembly in Living Cells: Insights from Dynamic and Single Molecule Microscopy

**DOI:** 10.3390/v11010072

**Published:** 2019-01-16

**Authors:** Kaushik Inamdar, Charlotte Floderer, Cyril Favard, Delphine Muriaux

**Affiliations:** IRIM, CNRS UMR9004, CNRS & University of Montpellier, 34293 Montpellier, France; kaushik.inamdar@irim.cnrs.fr (K.I.); charlotte.floderer@orange.fr (C.F.)

**Keywords:** HIV assembly, SMLM, dynamics

## Abstract

The HIV-1 assembly process is a multi-complex mechanism that takes place at the host cell plasma membrane. It requires a spatio-temporal coordination of events to end up with a full mature and infectious virus. The molecular mechanisms of HIV-1 assembly have been extensively studied during the past decades, in order to dissect the respective roles of the structural and non-structural viral proteins of the viral RNA genome and of some host cell factors. Nevertheless, the time course of HIV-1 assembly was observed in living cells only a decade ago. The very recent revolution of optical microscopy, combining high speed and high spatial resolution, in addition to improved fluorescent tags for proteins, now permits study of HIV-1 assembly at the single molecule level within living cells. In this review, after a short description of these new approaches, we will discuss how HIV-1 assembly at the cell plasma membrane has been revisited using advanced super resolution microscopy techniques and how it can bridge the study of viral assembly from the single molecule to the entire host cell.

## 1. Introduction

The human immunodeficiency virus (HIV) is an enveloped single stranded dimeric (+)RNA virus belonging to the family *Lentiviridae*. HIV is the causative agent of acquired immuno-deficiency syndrome (AIDS), estimated to have infected 70 million people since the first reported case, with 37 million people currently infected worldwide (source WHO 2017). However, since the introduction of protease inhibitors in HIV treatment, AIDS is considered a chronic disease. The HIV replication cycle has been widely studied. Briefly, HIV-1 infects mainly CD4+ T-lymphocytes by binding to the primary CD4+ receptor via its envelope (Env) protein and fuses by binding to the co-receptors, CCR5 or CXCR4, depending on the cell tropism. Post-fusion, uncoating of the virus takes place in the cytoplasm followed by reverse transcription and nuclear import of the viral DNA. The viral DNA integrates into specific sites in the host cell genome and then subverts the host cell machinery to transcribe and translate the viral genes into genomes and proteins, which are then trafficked to the membrane and assembled first into immature virions. The viral RNA genome is then packaged as a dimer into the forming particle. Concomitant to the particle release, post-cleavage of the structural Gag proteins by the viral protease occurs to give rise to its mature form, where a large structural rearrangement of the viral particle renders it infectious (for general reviews, see [1,2]).

The HIV-1 Gag structural protein is a 55 kDa polyprotein containing four domains; matrix (MA), capsid (CA), nucleocapsid (NC), and p6, in addition to two small spacer peptides, SP1 and SP2. Approximately 3000 Gag polyproteins assemble to form a single immature capsid shell [3]. Mutational analyses have revealed that only three domains of Gag (MA, CA, and NC) are required for immature particle assembly, whereas the fourth domain, p6, is required for budding and release by recruiting the endosomal sorting complexes required for transport (ESCRT) machinery host cell factors [4]. Membrane targeting of Gag requires the N-terminal myristate of MA, as well as residues in MA that form a basic patch (the highly basic region, HBR) which interacts with the acidic head groups of phospholipids at the inner leaflet of the cell plasma membrane (PM), especially including the phosphatidylinositol 4,5-bisphosphate (PI(4,5)P2) [5,6,7,8,9,10,11,12,13]. CA contains residues that form critical Gag–Gag interactions and NC is required for viral genomic RNA packaging, as well as non-specific interactions with RNA [14], and is essential for particle assembly [15]. Indeed, Gag is the only viral protein required for assembly and release of immature viral particles in cells, although production of infectious viruses requires other viral proteins, the Env proteins and the genomic RNA.

## 2. What Is Known about HIV-1 Assembly from Classical Microscopy?

The mature HIV-1 particle is 80–150 nm in diameter size [16,17,18] and thus the assembly site is below the resolution limit of conventional optical microscopy methods. Electron microscopy has long been a standard to study HIV-1 assembly and the particle structure [19]. Several studies have used electron microscopy to successfully demonstrate Gag multimerization at the plasma membrane, membrane curvature at the assembly site and viral budding. The role of cellular factors, such as the ESCRT machinery in HIV-1 particle release, has also been elucidated by this method [4,20]. Scanning and transmission electron microscopy has offered great insights into the formation and morphology of viral particles, providing evidence for electron-dense Gag layers underneath the plasma membrane and nascent particles connected to the cell surface by a thin stalk (for example see [21]). Further advances such as cryo-electron tomography have dropped the resolution to a few angstroms and allowed visualisation of viral protein complex structures close to the atomic level [22,23,24]. However, despite its stellar contribution to our understanding of HIV-1 particle structures, electron microscopy does not allow for study of the dynamics and the kinetics of nanoscale events taking place during virus assembly at the host cell plasma membrane.

The fluorescence microscopy advances, coupled with genetically encoded fluorescence proteins or external fluorophores added on live cells, has offered new insights into the dynamics underlying HIV-1 assembly. In the past decade, the introduction of a small Cys rich tag into the viral Gag protein enabled dynamic fluorescent imaging of Gag, using the membrane-permeable biarsenical compounds, FlAsH and ReAsH, in model cell lines and in macrophages [25,26]. Since HIV-1 assembly occurs at the cell plasma membrane, most of the improvement in deciphering this process was achieved by total internal reflection fluorescence microscopy (TIRF-M). The kinetics of HIV-1 Gag assembly and particle release could then be visualized thanks to fluorescent tagged Gag proteins, for the first time, in a transfected adherent model cell line [27]. Using live two color TIRF microscopy, the timing of the association of Gag with the genomic RNA at the cell membrane [28,29] could be observed for the first time, and thereafter with the ESCRTs for particle budding [30,31]. Using multicolor frame-by-frame alternating TIRF/wide-field imaging, Ivanchenko et al. pointed out a possible pre-association of several cytosolic Gag molecules with the genomic RNA [32]. It was suggested that there is a gradual formation of Gag multimers at the plasma membrane from the cytosolic pool [28,32]. Later on, advanced dynamic confocal microscopy techniques, such as raster image correlation spectroscopy (RICS) [33], confirmed the existence of the Gag-RNA oligomers in the cytosol that could be the precursors of assembly [34].

The recruitment of the ESCRT proteins, needed for bottle neck membrane scission of the viral particle, was also shown to be sequential [30,31]. Tsg101 is first recruited with Gag at the assembly site [30], interacting concomitantly with p6, and the NC domain of Gag [35], and persisting, like another ESCRT protein, ALIX [31,36], while CHMP and Vps4 proteins come later and are transient [30,31,37]. The entire process is no longer than 20 min [27,32] and seems to be cell type independent [38]. The scission and release process has been extensively reviewed in the past 5 years [36,39] but clarity still eludes us regarding the time interval between the 5 min of assembly/multimerisation of Gag proteins at the cell membrane and the late transient CHMP/Vps4 recruitment (Figure 1). Most of the above results are schematized in Figure 1.

Although conventional fluorescence microscopy methods such as wide-field or confocal microscopy have succeeded in elucidating many features of HIV-1 assembly in cells and continue to shed a light on its dynamics, they are severely confined by the diffraction limit (d~250 nm), preventing access to the molecular details of the assembling virus (*d* < 150nm).

## 3. Advanced Super Resolution Microscopy Shed a Light on HIV-1 Assembly

In these last two decades, far field optical microscopy has overcome the old resolution barrier, the diffraction limit, enabling super resolution visualization of previously invisible molecular details in biological systems. Since their conception, super-resolution imaging methods have continually evolved and one could expect that they will allow imaging cellular structures in living systems in three dimensions, with multiple colours, at the nanometer scale resolution, even though it remains very challenging. These techniques have been used to answer questions involving organization, interaction, stoichiometry, and dynamics of individual molecular building blocks and their integration into functional cellular machineries [41]. Super resolution microscopy (SRM) techniques can be divided into two categories. The first one is based on shaping the illumination light, such as (saturated) structured illumination microscopy (SIM) and stimulated emission depletion microscopy (STED). STED involves selective deactivation of fluorophores combined with this illumination shaping to narrow the emission spot down to 20–30 nm [42,43]. The second method is based on single-molecule detection and localization (often called single molecule localization microscopy, SMLM) taking advantage of the fluorophore ability to blink or to be photo-converted [44]. In this category, are found stochastic optical reconstruction microscopy (STORM) [45] and photo-activation localization microscopy (PALM) [46]. These approaches rely upon the acquisition of a set of several thousand images with each image having a random subset of fluorophores stochastically fluorescent at a given time point. By detecting each of single fluorophores and localizing precisely their center of mass, single molecule localizations from each of thousands of images can be obtained. These localizations are then used to reconstruct the final image with single molecule localisation precision (for a review see [47]). This localization precision is inversely proportional to the number of photons detected for each single molecule. Therefore, using very bright fluorophores, 10 to 20 nm precisions can be achieved.

Benefits from these nanoscale microscopy techniques have been rapidly applied to the study of immature HIV-1 Gag(i)mEOS2 assembly (see Figure 2 for an illustration). In fact, HIV-1 Gag was one of the first proteins to be visualized by PALM SRM [46] showing different cluster sizes of Gag, tagged with mEOS, at the cell plasma membrane of adherent model cell lines. Single molecule coordinate based analysis has shown the existence of three different types of Gag clusters in adherent cells: small random clusters (inferior to 50 nm diameter), clusters of defined size corresponding to assembling sites (between 50 to 130 nm) and large patchy aggregations of Gag corresponding to fully assembled structures (~140 nm diameter) [50]. Because the introduction of a protein tag into Gag could play a role, Gunzenhäuser et al. compared two photo-activable tags, mEOS2 and tdEOS, with respect to Gag assembly [51]. The results show that Gag tagged with tdEOS forms unusually large clusters as compared to Gag-mEOS2, which in turn forms clusters well within the range for HIV-1 assembly sites. Their data suggests that the addition of protein tags may very well change the nature of Gag assembly. The larger tandem dimeric tdEOS possibly disrupts the regular hexameric Gag lattice structure in the VLPs, changing Gag organization. Introduction of fluorescent protein tags has previously been known to perturb Gag VLP structure. Indeed, electron microscopy studies have shown abnormalities in VLPs formed by Gag-GFP molecules [52,53] and in another study, HIV-1 tagged with an internal GFP in the C terminus of MA exhibited poor infectivity in the absence of helper untagged Gag molecules [54]. It is thus critical to add a certain quantity of untagged Gag along with the fluorescently tagged molecule. However, Hubner et al. shows that a Gag tagged internally with GFP, between the CA and the MA domains, flanked by protease cleavage sites, can facilitate native processing of the Gag polypeptide. This Gag(i)GFP was able to mimic native Gag in terms of localization and assembly dynamics [55]. Fluorescent protein tags appear as essential tools to study viral assembly in host living cells. However, choosing the right fluorophore requires consideration of its intrinsic properties and its location into the targeted protein. Genetically encoded fluorescent tags and their properties have been reviewed in [56], and their applications and limitations for live cell imaging in [57,58]. Furthermore, recently, a comparative study of fluorescent tag properties for temporal or spatial fluorescence fluctuations was reported for quantifying protein oligomerization in living cells [59]. Several fluorescent proteins have been engineered from dimeric or tetrameric jellyfish or coral derived proteins and they may thus retain some residual tendency to aggregate, which may cause significant effects on the dynamics of protein–protein interaction studies. This is especially true when studying viral assembly: some FP proteins could thus reduce or increase HIV-1 Gag protein oligomerization properties and affect VLP biogenesis. In addition, fluorescence imaging in live cells carries with it an additional risk factor of phototoxicity, such as photobleaching, photodamaging, or the creation of photoreactive oxygen species due to the fluorophore and illuminations. This is especially evident in cases of photoactivable fluorescent proteins (PAFPs), such as EosFP, which require activation in the near-UV range of the spectrum. Strategies to reduce phototoxicity, by limiting the illumination to the focal plane and controlling phototoxicity effects have been reported [60]. In addition, recently, new labeling protocols based on fluorogenic systems [61] and biosensors in general have been developed for measuring acute spatiotemporal events in living cells (review in [62]). Thus, the choice of a suitable tag, the experimental conditions and the illumination plane are critical as it has clear implications on the results and interpretations.

Apart from the Gag protein self-assembly properties, the formation of an infectious HIV-1 particle involves the viral RNA genome (gRNA), other viral proteins, such as the envelope transmembrane glycoproteins (Env), and some host cellular proteins, such as ESCRT (reviewed in [15,20]). Most of the SRM studies on HIV-1 assembly have focused on HIV-1 Gag and Env clusters at the cell membrane. Dual color PALM/dSTORM and other dSTORM studies have illustrated the importance of the CT domain of Env in HIV-1 Gag assembly [63,64]. Mutations in the CT domain of Env resulted in loss of specificity of Env incorporation, as revealed by a coordinate based distance distribution analysis [63]. In addition, image-based morphological cluster analysis showed that Env-WT clusters are larger in cells expressing all HIV-1 proteins (except Nef) as compared to Env-WT only transfected cells, while it was not the case for EnvΔCT. This enhanced clusterization of Env trimers in the presence of Gag was also observed by Roy and coworkers [64]. A few SRM studies have gone further and looked at cell-free virions to detect viral proteins and their positioning within the particle. A STED study in 2012 [65] revealed the distribution of Env on the virus surface which was dependent on the maturation process as well as on the Env CT domain.

Studies regarding host cell factors in HIV-1 assembly have mainly focused on the ESCRT machinery. Van Engelenburg et al. looked at relative distribution of ESCRT subunits within the budding particle by iPALM (interferometric PALM, a 3D SRM technique, see [66]) and correlative electron microscopy [67]. They showed the initial scaffolding of ESCRT subunits CHMP2A, CHMP4B and TSG101 within the viral bud followed by levels of CHMP2A decreasing significantly relative to Tsg101 and CHMP4B upon virus abscission. Apart from shedding light on the spatial distribution of ESCRT subunits within the particle, thanks to iPALM/correlative SEM, the differential incorporation of CHMP2A points to a distinct dynamic among the ESCRT subunits preceding viral abscission. TSG101 revealed a small cytosolic pool punctuated by localizations within the Gag lattice of assembling particles. Using two-color two dimensional super-resolution imaging, the ESCRT machinery (TSG101, ALIX, and Chmp4b/c proteins) was observed to be positioned at the periphery of the nascent virions, with TSG101 closer to the assembled Gag than ALIX, Chmp4b, or Chmp4c [30], certainly due to its direct interaction with NC and p6 [35]. Finally, Prescher and collaborators showed that the membrane scission process is driven from inside the HIV-1 budding neck by ESCRT-III protein assemblies including CHMP4B andCHMP2A as well as TSG101 and ALIX which were also located in clusters similar to the dimension of the neck [68].

All these studies show that SRM can now help in deciphering the respective positions of the different molecules involved in viral particle assembly in cells. Moreover, correlative iPALM and SEM studies by Van Engelenburg et al. [67] and Pedersen et al. [69] highlight the possibilities of SRM shedding light on a more resolved HIV-1 particle structure.

One of the main advantages of SRM over electron microscopy is its ability to observe, in real time and in living cells, ongoing processes at the nanoscale level, making it a technique of choice for studying virus assembly. However, most of the SRM studies cited above, barring a few [64], were limited to fixed cell imaging. In the last five years, many efforts have now been put into deciphering the real-time molecular events occurring in living cells during the HIV-1 assembly and budding process.

## 4. Towards a Real Time Molecular Description of HIV Assembly in Living Cells

The first attempt to study the spatiotemporal aspect of Gag assembly in living cells at the single molecule level was conducted with single particle tracking PALM (sptPALM) [70]. By extracting the apparent diffusion coefficients (D) of each single Gag molecule from their mean square displacement (MSD), Manley et al. observed a strong decrease in Gag mobility that was correlated with the presence of Gag clusters at the cell plasma membrane (see Figure 3). Later on, a comparison of apparent diffusion coefficient distribution highlighted the role of the NC-RNA binding domain of Gag in HIV-1 assembly in adherent cells [71] or in CD4 T-cells [49]. Recently, studies by Floderer et al. [38,48] in T cells, and Yang et al. [72] in adherent cells, used sptPALM to track Gag molecules and to confirm the important role of the gRNA and the NC domain of Gag in the generation of Gag clusters at the plasma membrane of the host cell for efficient HIV-1 assembly. Floderer et al. [38,48] showed that gRNA in the context of a provirus acts as a spatiotemporal coordinator of the membrane assembly process, rather than only serving as an attractor for Gag molecules to the assembly site. By monitoring the trajectories of single HIV-1 Gag molecules in the vicinity of assembly sites, Floderer and collaborators [38,48] have shown the existence of directed motions towards these assembly sites. It is currently not clear or elucidated what these Gag directed motions towards the assembly site are; it could reflect Gag multimerization on the viral RNA, Gag–Gag interactions, Gag–host cell factor interactions or others. Using different Gag mutants and live SRM, several studies have revealed a major role of the NC–RNA interaction in coordinating HIV-1 Gag assembly at the cell plasma membrane [48,71,72]. Indeed, it was observed that, contrary to the hypothesis that gRNA only functions to drive formation of low-order Gag multimers, there was a crucial dependency of the assembly process at the cell membrane on the NC–RNA interactions.

Live SRM requires big data analyses that can be monitored in different ways. Yang et al. [58] classified the Gag trajectories using their mean square displacement and the deviation from the purely diffusive (random) motion. They used a α > 1.3 exponent to consider directed motions from the generalized description of the MSD (see Figure 3b–d for detailed explanation). In the Floderer et al. approach, a Langevin description of Gag motions was used to rigorously disentangle the attractive (directed) and the random (Brownian) part of the overall Gag motion. Coupling this with Bayesian inference and Voronoi tessellation [73], Floderer et al. established Gag diffusion and attraction energy maps and monitored their fluctuations during the entire assembly and budding process (30 min) at the nanoscale level (see Figure 3e). Interestingly, their results show that the attraction strength sensed by single membrane Gag proteins at the vicinity of the assembly sites was not dependent on the nature of the RNA and was equivalent to an attractive energy of 3 to 4 kT (2–3 kcal/mol). Furthermore, it was revealed that the Gag–Gag interaction mediated by CA–CA dimerization represented only a third of Gag attractive energy, while the NC–RNA platform represented two thirds [48].

Measurement of apparent binding energies during viral assembly in living cells is opening new perspectives and allows comparison of live cell measurements with nicely tuned and controlled in vitro systems or in silico molecular simulations. For example, coarse grained modeling recently predicted that the presence of gRNA lowered the necessary SP1-CA hexameric interaction energy down to 3 kT to achieve correct HIV-1 assembly [74]. This energy value is indeed close to the Gag attractive energy found in Floderer et al. [48]. For the sake of comparison with another virus, an in vitro experimental approach found that the binding energy of icosahedral viral capsid subunit to its RNA genome was ~7 kT [75]. The capacity to directly measure the energy sensed by Gag during HIV-1 assembly using single molecule microscopy opens an incredibly large field of research that can now link coarse grained modeling and in vitro quantitative measurements and experiments in living cells.

Env incorporation into HIV-1 virions during particle assembly was recently studied by Buttler et al. [76]. Using iPALM, they showed that Env was incorporated in preformed Gag lattices to the neck of the assembling virions, as determined by the Env angular distribution on the surface of cell-associated virus. In parallel, a role of the C-terminal domain of Env in its trapping at the assembly site was elucidated, using sptPALM, by comparing the diffusion of the wt-Env with an Env deficient in the C terminal tail (EnvΔCT). This nanoscale visualization of the budding site highlights a role of Gag in Env incorporation. Finally, Env mobility in the lipidic membrane of released virion was also studied using another SRM technique, i.e. scanning STED-FCS. This SRM technique uses the fluorescence fluctuations of several molecules to measure their motion in each STED voxel of the line scanned [77]. Chojnacki et al. reported that Env mobility was really low (D = 10^−3^ µm^2^/s) in the virions and constrained by its C-ter domain [78]. This Env mobility was doubled upon virus maturation. Interestingly, the Env diffusion coefficient found by Chojnacki et al. in HIV-1 particles [78] is a hundred times smaller than the one observed at the plasma membrane of virus producing cells [76,78], leaving an open window regarding the exact process and dynamics of Env incorporation during HIV-1 assembly.

In conclusion, advanced super resolution microscopies are nowadays opening incredible avenues in which to revisit some aspects of virology, looking at the extreme small (below 100 nm) with the possibility of coupling it to live cell imaging. This will enable one to decipher the virus life cycle in the host cell at the level of the single molecule and spatiotemporally offering a 3D + t view of the events, which was not possible 10 years ago.

## Figures and Tables

**Figure 1 viruses-11-00072-f001:**
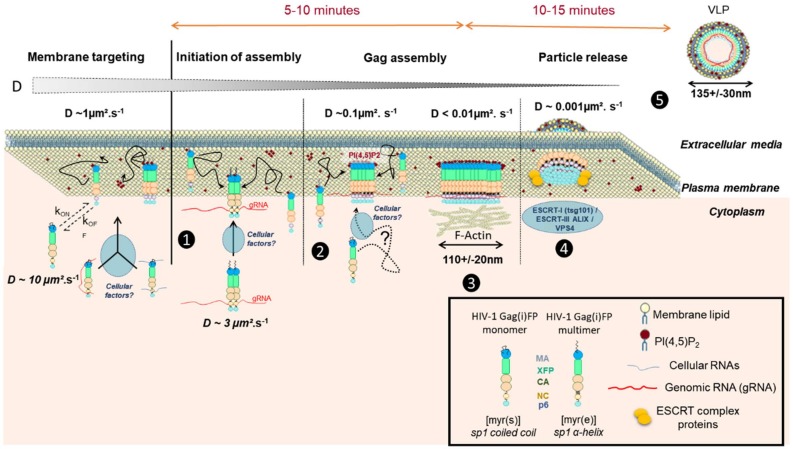
Scheme of HIV-1 Gag protein dynamics in the cytosol and at the cell plasma membrane during the process of viral particle assembly. From the left to the right: (1) The immature HIV-1 Gag(i)FP tagged protein is synthesized in the cytosol, where one can measure a diffusion coefficient of 3 µm^2^/s for cytosolic Gag probably as a dimer and already in association with the genomic RNA (labelled as gRNA in the figure) (from [32,34]). It is not known clearly if any cellular factors are involved at that stage or simple diffusion of Gag permits its targeting to the cell plasma membrane. (2) Once at the cell membrane, Gag interacts preferentially with acidic lipids, such as PI(4,5)P_2_, and anchors the cell membrane through its myristate domain. Then Gag can dynamically get back to the cytosol (measurable D~3 µm^2^/s) or pursue its multimerization on the viral RNA and at the inner leaflet of the cell PM (measurable D~1 µm^2^/s) (from [38]), thanks to concomitant NC-gRNA and the MA-lipid interactions. (3) After 5 min of assembly, i.e. Gag multimerization with incorporation of new Gag molecules, 10 more minutes are then required before the recruitment of the ESCRT machinery (4) (review in [36]), with a very low diffusion coefficient of Gag at the budding site (D~0.01 µm^2^/s)(from [38]). During this phase (3), the cortical actin network could play a role in stabilizing Gag multimers at the cell membrane (from [40]). (4) Then after ESCRT recruitment, by the NC and p6 domain of Gag, (5) the immature viral like particle (VLP) is released.

**Figure 2 viruses-11-00072-f002:**
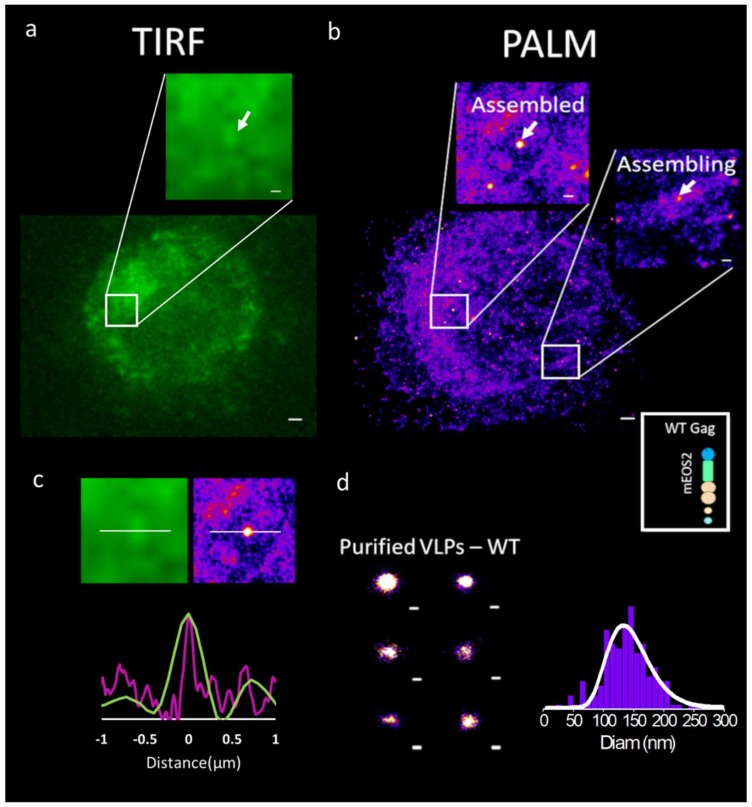
A direct imaging comparison of wild-type HIV-1 Gag(i)mEos2 in Jurkat T cell using (**a**) classical TIRF microscopy and (**b**) photo activated localisation microscopy (PALM). Insets are showing assembling (part b, right inset) or assembled structures (part a, and b left inset), (**c**) Direct comparison of fluorescence intensity profile (green, TIRF image), or number of localizations (purple, PALM image) along the x direction (white line) crossing an assembled viral like particle (VLP) shows enhanced resolution in the case of PALM allowing for direct measurement of VLP sizes. (**d**) Purified HIV-1 immature Gag VLPs as seen by PALM (left). With the help of PALM microscopy, it is possible to measure the diameter of each single VLP and establish a distribution of it (graph, right). The average diameter found here is ~140 nm. Scale bars are 0.5 µm for a and b, 0.2 µm for b inserts and 0.1 µm for d (adapted from [48,49]).

**Figure 3 viruses-11-00072-f003:**
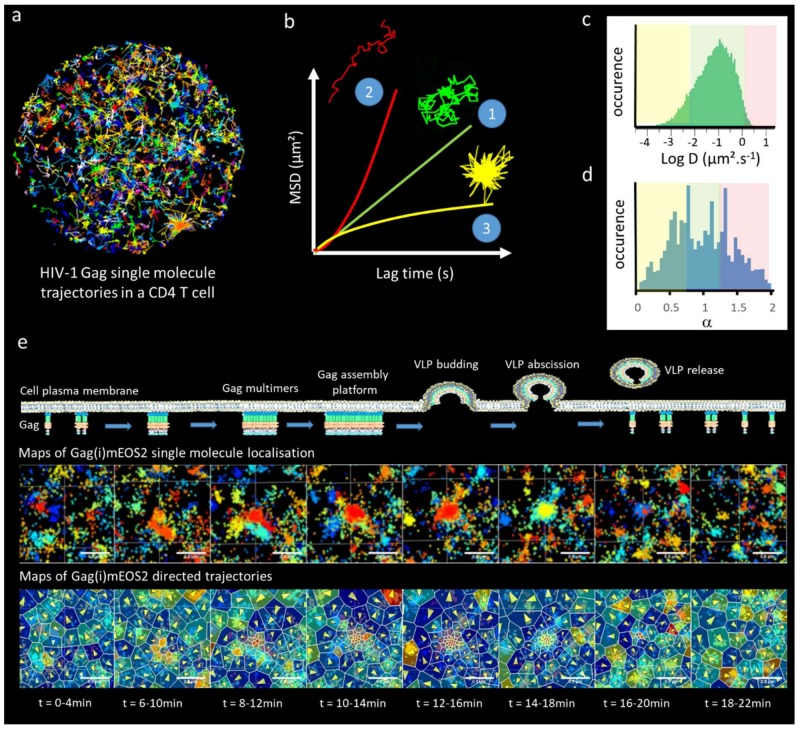
Single Gag molecule dynamics during HIV-1 assembly. (**a**) Trajectories of numerous single Gag observed by TIRF-sptPALM Jurkat T-cells. (**b**) 3 classes of trajectories can be isolated, corresponding either to pure Brownian motion (1, green track), directed motion (2, red track) confined or trapped motion (3, yellow track). By plotting the mean square displacement (MSD) of the single molecule as a function of time, each type of motion could be theoretically distinguished. MSD(t) curves as a function of the motion type (Brownian, green, Directed, red, Confined, yellow) could be non-rigorously approximated by the following relation MSD(t) = Dt^α^ where D is an apparent diffusion coefficient and α is either less than 1 (confined motion) or more than 1 (directed motion), with a special case where α = 1 (Brownian motion). (**c**) Due to noise in the MSD reconstruction it is often difficult to distinguish between confined, Brownian and directed motions, therefore a simple way to analyze the curves is to fit the MSD(t) with the simple and linear MSD(t) = Dt equation that usually leads to a broad distribution of apparent diffusion coefficients (distribution obtained from the trajectories observed in part (**a**)). These apparent diffusion coefficients could for example be arbitrarily divided in each class of motion (yellow, confined, low apparent diffusion coefficient, green Brownian, average apparent diffusion coefficient, red, directed, high apparent diffusion coefficient). (**d**) A less arbitrary method is to fit all the MSD(t) curve with the MSD(t) = Dt^α^ approximation and take the benefit of the numerous noisy curves to establish a statistically relevant distribution of motion, allowing one to qualitatively classify the type of motion observed in the cell. This technique has been used to observe directed motion close to the assembly site in [72]. (**e**) A more rigorous analysis aims at disentangling the part of Brownian and directed motion in the trajectory using the Langevin equation of motion. This approach coupled to spatial tesselation, allows one to generate maps of directed motions containing the average direction as well as the strength of the attraction force generating these directed motions. By coupling this information to the changes in the density of single Gag localization over time, we monitored the attractive energy sensed by each single Gag molecule in the vicinity of the HIV-1 assembly site in host CD4 T cells [48].

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
