# Peer review of "Monitoring HIV-1 Assembly in Living Cells: Insights from Dynamic and Single Molecule Microscopy"

_viruses, 2019, doi:10.3390/v11010072_

Reviewer 1 Report

This is a review of the manuscript by Kaushik et al which intends to provide a review of live cell imaging and super resolution microscopy of HIV budding. Overall, I have found the review interesting however it would tremendously benefit the field if two major issues were also discussed.

Major issues not discussed:

The first issue to a large extent omitted by the authors is the effects of phototoxicity and its role in limiting the live cell imaging techniques.

The second issue is a proper discussion on fluorescent tagging of biological molecules and the effects of the tags on the observed kinetics. 

Minor points:

Line 10, the sentence starting with researchers does not read well and needs a rewrite. The same applies to the rest of the abstract.

First paragraph of introduction is a bit inaccurate. For example its correct that HIV has a single strand RNA genome but the fact that its positive stranded and there are two strands embedded in each virion is omitted. The reference for 50% mortality is also out of date, since 1990’s after introduction of protease inhibitors AIDS has transitioned to a chronic disease.  This paragraph needs to be re written to more accurately reflect the virology of HIV.

While incorporation of TSG101 is discussed by the authors, the incorporation of ALIX is missing from the text.

Lines 38-39. It has been recently shown that p6 is not required for release of HIV virions and rather it affects kinetics of release with respect to protease activation. A broader discussion of the role of p6 may be appropriate since kinetics seem to be forefront on the authors mind.

Line 180-216, There is a very detailed comparison of the work performed by authors in two manuscripts with that of Yang et al., while I find this comparison very interesting, it lacks much context in its presentation, for example how the free energies were calculated. There is suddenly too much detail in this paragraph which does not match with the overall depth of the review. Also since this is a review article, the authors may wish to write the whole paragraph in third person.

Author Response

Answers to Reviewer #1:

This is a review of the manuscript by Kaushik et al which intends to provide a review of live cell imaging and super resolution microscopy of HIV budding. Overall, I have found the review interesting however it would tremendously benefit the field if two major issues were also discussed.

Major issues not discussed:

The first issue to a large extent omitted by the authors is the effects of phototoxicity and its role in limiting the live cell imaging techniques.  

The second issue is a proper discussion on fluorescent tagging of biological molecules and the effects of the tags on the observed kinetics. 

We thank the review 1 for these remarks that we totally agree with. In order to lessen this, we included a whole section on phototoxicity and fluorescent tags effects, from lanes 175-193, in addition of references of recent reviews on these subjects.

Minor points:

Line 10, the sentence starting with researchers does not read well and needs a rewrite. The same applies to the rest of the abstract.

We thank the reviewer for this remark. We have rewritten the abstract and we hope it now reads well. We also paid more attention to English corrections.

First paragraph of introduction is a bit inaccurate. For example its correct that HIV has a single strand RNA genome but the fact that its positive stranded and there are two strands embedded in each virion is omitted.

We have corrected this by adding “dimeric” in the sentence “HIV is an enveloped single stranded dimeric (+)RNA virus” lane 25 and a new sentence in lane 36 in the introduction section.

The reference for 50% mortality is also out of date, since 1990’s after introduction of protease inhibitors AIDS has transitioned to a chronic disease.  This paragraph needs to be re written to more accurately reflect the virology of HIV.

Regarding this remark, the paragraph has been changed from lanes 26 to 29 and the sentence “However, since the introduction of protease inhibitors in HIV treatment, AIDS is considered as a chronical disease.” was added in the introduction section.

While incorporation of TSG101 is discussed by the authors, the incorporation of ALIX is missing from the text.

ALIX was re-introduced along several lines in the text with some associated references: see lanes 91, 221-227.

Lines 38-39. It has been recently shown that p6 is not required for release of HIV virions and rather it affects kinetics of release with respect to protease activation. A broader discussion of the role of p6 may be appropriate since kinetics seem to be forefront on the authors mind.

 Thanks for this remark but we consider that this review is not at all about discussing the role of p6 in HIV release; we only refer to the p6 domain of Gag in a general manner.

Line 180-216, there is a very detailed comparison of the work performed by authors in two manuscripts with that of Yang et al., while I find this comparison very interesting, it lacks much context in its presentation, for example how the free energies were calculated. There is suddenly too much detail in this paragraph which does not match with the overall depth of the review. Also since this is a review article, the authors may wish to write the whole paragraph in third person.

This section has been rewritten using the third person. We tried to diminish the weight of the detailed comparison of the studies with respect to the other sections. Nevertheless, we kept details regarding the energies because we wanted to emphasized the power of this type of spt-PALM data analysis and the possible connection of the information these analyses provide to the one that could be obtained from coarse grained molecular dynamics. We did not get into details of how the energy are monitored because we thought this will be to technical. The energy measured in live cells at the assembly site by spt-PALM reflects an attractive potential which nature remains to be determined. Nevertheless, we found interesting to illustrate the fact that this attractive potential is within the range of the one used in coarse grained numerical simulations (Pak et al.) to simulate HIV-1 assembly, or the binding energy experimentally measured during in vitro assembly of icosahedral viruses.

Reviewer 2 Report

Inamdar, K. et al contribute an important review to the virology field highlighting the results and future potentials of single molecule and superresolution technologies to the study of viruses and viral infections. The authors do a good job of covering the broad applications of these single molecule techniques to the intricate aspects of HIV-1 assembly specifically. While this the subject matter is novel, modern, and exciting from the perspective of virology in general. This review suffers from major structural organization issues, vague and non-quantitative descriptions of the results of cited studies, a somewhat biased focus on the author’s research. This reviewer would, however, consider accepting this manuscript for publication if these major and specific issues are addressed.

Major points:

The manuscript reads as if three separate authors wrote their own subsections, put it together as one manuscript, and then didn’t re-write the entire review to have more cohesion. This must be addressed by organizing the subsections to have logical flow between the various aspects of HIV-1 assembly. Might I suggest, focusing on one assembly process (as example RNA or Env) instead of interweaving this review with the various assembly steps.

Section 3, paragraph 1 is very unprofessional and biased by the authors. The authors contrast what appears to be a competitive study, published earlier than their own, on the potential forces of RNA on Gag during HIV-1 assembly. The authors crudely state that Yang, et al are “non-rigorous” in their MSD analysis. While all single particle tracking analysis methods are limited and flawed in particular ways, to say that MSD analysis, which has been used in 1000’s of single molecule imaging/tracking studies, is non-rigorous is untrue. It appears to this reviewer that Yang’s et al use of alpha exponent > 1.3 is a conservative cutoff for single molecules which are undergoing anomalous superdiffusion (defined as alpha > 1). For the authors to then claim that their analysis methods are far superior only complicates the review and is too technical for a general virology audience. While the authors contribution to inference of attraction energies between Gag-vRNA is certainly exciting, I think this entire section can be written much more generally and less biased to capture the strengths of multiple approaches to interrogate the role of vRNA on Gag assembly.

Instead of using figures from one laboratory, can permissions be requested to highlight the multitude of other beautiful superresolution and imaging experiments cited in this review? This manuscript is very biased to the authors work.

Section 1 could use an image for the audience less familiar with novel imaging technology to contextualize the imaging approach to resolving single virus sites, etc.

Citation 34 cannot be used as it is a reference to a non-peer reviewed pre-print on bioRxiv and seems like the identical study published in reference 45.

It has been demonstrated that actin may not play a direct role in the HIV-1 assembly process. The authors must discuss and cite: Rahman J. Virol 2014, 88(14). Which thoroughly shows that direct actin perturbing drugs do not affect particle assembly and release significantly. The language in the manuscript (line 104 and elsewhere) is too strong in support of actin in this process, with only one reference, Thomas et al. were also looking at the Rac/Rho signaling pathways and not more directly at actin as in Rahman et al. Given this competing study, the role of actin in HIV-1 assembly, budding, and release is at best controversial, but not certain as has been written.

Section 2 does not adequately quantify the relative resolution improvement of each superresolution imaging technique. To say with such certainty that STED improves the resolution 10-fold is not accurate (give a range of fold improvements to help with reader expectations of these techniques; also us “-fold”). Also, a qualifying sentence supporting the role of bright probes (photon statistics) for other techniques such as STORM/PALM as dictating the resolution would give a virology audience more appreciation for the requirements for these techniques. I understand if this makes the manuscript more technical, but give it a try.

The manuscript must be copy-edited to capture grammatical mistakes that change the meaning of sentences.

Specific points:

Figure 1: Text too small, increase font size in figure and bold.

Lines

38-39: Use references from Freed, E.O. Virology 251, 1998, 1-15. To cite as examples of this topic

36: Should state: 3000-5000 Gag polypeptides as concluded by the reference included (keep reference)

49: Provide a citation for this particle range

60: Add reference for Carlson, L.A. et al 2010. Plos Pathog. 6(11) and highlight cyro-EM on think cellular sections with viral budding

71-72: Why is the 7SL study important it seems to detract from the Gag vRNA focus of the review (simply, this reference is out of place)

76: Jouvenet et al 2008 Nature (ref 24) should be used here since this was the first example.  Remove reference 25 from this sentence.

77-78: Remove reference 28 and add the earlier reference looking at HIV and ESCRT via TIRF: Jouvenet et al NCB, 2011, 13(4)

81-85:  This is a fragmentary sentence and doesn’t make any logical sense.

110-130: Primary citations should come first and they reference the readers to a review on the subject at the end of the paper (section 2)

123-125:

Figure 2: There is not inset for comparison between resolution improvements in a versus b. Can this improvement be quantified for the reader (simple linescan)?

144-147: The authors must cite predecessor papers showing that the size and location of the fluorescent probe on the Gag polypeptide has been shown to perturb lattice assembly. Hubner et al. 2007, J Virol. 81(22), who was the first to create the internal MA-CA FP fusion for gag with GFP was careful to show that all the molecules of Gag in the cell cannot be tagged with the fluorescent protein. It would be important to make clear that doping with untagged Gag is critical to not perturbing the HIV assembly process. Are the images provided in reference 45 and in this manuscript performed in the presence of untagged Gag at suitable rations (>1:3 tagged to untagged)? This is also highlighted in papers from Jouvenet et al. 20xx.

155-157: This sentence is not clear and doesn’t capture ref 50 conclusions accurately.

157-159: distinguish what present is relative to (e.g. is present at assembly sites?)

165: Reference VerPlank, L et al. PNAS 2001 for discovery of Tsg101 as p6 interactor.

167-168: This sentence is very vague, what are the important details for the reader?

170: Remove the word “while” and replace with “as well”

160-171: The technologies used here must at least describe their performances. Why iPALM, it seems this reference uses 3D superresolution to directly measure the ESCRT positions, but it seems that Presher et al and Bleck et al used 2-dimensional imaging to infer the 3D position of ESCRTs. This is not clear.

175: This sentence doesn’t explain how this can open up possibilities. Of what? Very vague. This is not the first example of correlative iPALM and SEM (cited earlier in reference 52 line 162), what about this study improved possibilities?

177-181: No citations are used here to “bar a few”. This paragraph could be turned much more positive and highlight the strengths of the previous studies mentioned, but them transition to the extension to the temporal domains with superresolution imaging.

217: The term “aspects” is very vague. Be more specific: what aspects?

223: This sentence makes it seem like these studies are not informative. Remove the words “could now” and “possible”.

225: Define STED-FCS and perhaps highlight the strengths of this powerful technique (compare with SPT methods?).

226: Small is a very non-quantitative word. Word choice.

227: Use the word: “doubled”, “twice-increased” suggests that there are two doublings in steps.

227-228: The comparison of diffusion coefficiencet between Chojnacki et al. and Buttler et al. are not correct. STED-FCS measurements of Env on the cell surface (Chojnacki et al) are in good agreement with those of Env diffusion via SPT (Buttler et al.). The mobile fraction on cells was determined via spt whereas the diffusion on released particles via STED-FCS was shown to be 100-fold lower, but only in comparison to Env diffusion on the cell surface (prior to lattice incorporation).

Author Response

Answers to Reviewer #2:

Inamdar, K. et al contribute an important review to the virology field highlighting the results and future potentials of single molecule and super resolution technologies to the study of viruses and viral infections. The authors do a good job of covering the broad applications of these single molecule techniques to the intricate aspects of HIV-1 assembly specifically. While this the subject matter is novel, modern, and exciting from the perspective of virology in general. This review suffers from major structural organization issues, vague and non-quantitative descriptions of the results of cited studies, a somewhat biased focus on the author’s research. This reviewer would, however, consider accepting this manuscript for publication if these major and specific issues are addressed.

We thank the reviewer for his/her intensive work to review in details our manuscript. We tried to address all the different issues he/she addressed.

Major points:

The manuscript reads as if three separate authors wrote their own subsections, put it together as one manuscript, and then didn’t re-write the entire review to have more cohesion. This must be addressed by organizing the subsections to have logical flow between the various aspects of HIV-1 assembly. Might I suggest, focusing on one assembly process (as example RNA or Env) instead of interweaving this review with the various assembly steps.

Thanks to the reviewer, we agree on this point, thus, the review was somewhat re-written accordingly to have more cohesion. Especially, section 2 of the review was almost completely re-arrange and re-written. For a logical flow, we choose:

In section 1, to report what is known in HIV assembly from classical microscopy, thus addressing Gag or Gag/RNA interaction, then Env, then host cell factors involved in HIV particle budding, i.e. ESCRT proteins.

In section 2, we relate essentially, Gag assembly, then Env, and then ESCRT at assembly site from SRM data in fixed cells.

Finally, section 3 is reporting very recent articles showing how we can associate SRM and big data analysis of Gag motions in live cells in order to decipher HIV Gag assembly and VLP formation addressing Gag-gRNA interactions, then Env.

All the quantitative values extracted from all the articles are reported in figure 1.

Section 3, paragraph 1 is very unprofessional and biased by the authors. The authors contrast what appears to be a competitive study, published earlier than their own, on the potential forces of RNA on Gag during HIV-1 assembly.

The study was not published before as it was published the same year with very few months delay indicating that the study was driven during the same period of time, ie few couple of years ago.

The authors crudely state that Yang, et al are “non-rigorous” in their MSD analysis. While all single particle tracking analysis methods are limited and flawed in particular ways, to say that MSD analysis, which has been used in 1000’s of single molecule imaging/tracking studies, is non-rigorous is untrue. It appears to this reviewer that Yang’s et al use of alpha exponent > 1.3 is a conservative cut off for single molecules which are undergoing anomalous superdiffusion (defined as alpha > 1).

The sentence has been rewritten and the statement has been moderated. Indeed, by saying this, we wanted to point out a main issue in using MSDs in the case of spt-PALM. The trajectories are most of the time very short (4-15 time points) in spt-PALM, giving rise to very noisy MSD data, in contrary to classical single particle tracking. The noisy data, added to little time length of the tracking makes it very difficult to classify the nature of the trajectories using their deviation to normality, signed by a power law (anomalous sub or super diffusion).  Following this, anomalous super diffusion can arise from many different process that are not only ballistic plus diffusion processes (which will be analytically written as MSD(t)=4Dt+V^2t^2, in the case of a 2D Brownian motion+ballistic). Amongst these, Levy-flights are the most famous and are not associated with any source of attractive energy or directional motion. They could reflect a 2D diffusion with binding/unbinding to the membrane (as Gag could do in cells). In this case, the MSD will scale with a power law with time, the exponent being higher than one (See for example: Strange kinetics of bulk-mediated diffusion on lipid bilayers. Krapf D et al. .Phys Chem Chem Phys. 201618(18):12633-41 or Superdiffusive motion of membrane-targeting C2 domains. Campagnola G et al. Sci Rep. 2015 Dec 7;5:17721).

We did not consider going into all these details because we thought this would be clearly too technical for a general virology audience.

Finally, the 1.3 conservative cut-off value of alpha is arbitrary and really depend on the author’s choice. For example, in Neuronal Receptors Display Cytoskeleton-Independent Directed Motion on the Plasma Membrane. iScience. 2018;10:234-244. Taylor RD, et al., used a 1.51 cut off to discriminate between directed motion and Brownian motion. Therefore, there is no clear justification of this chosen values as cut-off, but an empirical choice. This is why we consider this approach to be less rigorous.

For the authors to then claim that their analysis methods are far superior only complicates the review and is too technical for a general virology audience. While the authors contribution to inference of attraction energies between Gag-vRNA is certainly exciting, I think this entire section can be written much more generally and less biased to capture the strengths of multiple approaches to interrogate the role of vRNA on Gag assembly. Instead of using figures from one laboratory, can permissions be requested to highlight the multitude of other beautiful superresolution and imaging experiments cited in this review? This manuscript is very biased to the authors work.

Accordingly to the reviewer’s remark, we wrote this section in a more general manner.

Indeed, we asked permission to another laboratory for the use of their data or images but it was a nightmare in order to get Journal authorization for the use of figures. Thus, we decided to only use general or summarized figures (figure 1) made in our lab in order to image our review.

Section 1 could use an image for the audience less familiar with novel imaging technology to contextualize the imaging approach to resolving single virus sites, etc.

We consider that this particular point is now better imaged in the new figure 2.

Citation 34 cannot be used as it is a reference to a non-peer reviewed pre-print on bioRxiv and seems like the identical study published in reference 45.

Ok. The citation in bioRxiv was removed.

It has been demonstrated that actin may not play a direct role in the HIV-1 assembly process. The authors must discuss and cite: Rahman J. Virol 2014, 88(14). Which thoroughly shows that direct actin perturbing drugs do not affect particle assembly and release significantly. The language in the manuscript (line 104 and elsewhere) is too strong in support of actin in this process, with only one reference, Thomas et al. were also looking at the Rac/Rho signaling pathways and not more directly at actin as in Rahman et al. Given this competing study, the role of actin in HIV-1 assembly, budding, and release is at best controversial, but not certain as has been written.

Because this review is not addressing or discussing the role of actin in HIV assembly, we removed it from the main text.

Section 2 does not adequately quantify the relative resolution improvement of each superresolution imaging technique. To say with such certainty that STED improves the resolution 10-fold is not accurate (give a range of fold improvements to help with reader expectations of these techniques; also us “-fold”). Also, a qualifying sentence supporting the role of bright probes (photon statistics) for other techniques such as STORM/PALM as dictating the resolution would give a virology audience more appreciation for the requirements for these techniques. I understand if this makes the manuscript more technical, but give it a try.

We thank the reviewer for these suggestions. This section has been rewritten accordingly by adding a sentence explaining the relation between the number of photons and the localization precision for SMLM techniques. The 10’-fold’ resolution improvement has been removed. We tried to avoid the use of “resolution improvement” terms. (see lines 130-144).

The manuscript must be copy-edited to capture grammatical mistakes that change the meaning of sentences.

The manuscript was re-written and hopefully mistakes disappeared.

 Specific points:

Figure 1: Text too small, increase font size in figure and bold.

This point was fixed.

Lines

38-39: Use references from Freed, E.O. Virology 251, 1998, 1-15. To cite as examples of this topic

This has been added.

36: Should state: 3000-5000 Gag polypeptides as concluded by the reference included (keep reference)

The number of Gag monomer has been changed accordingly.

49: Provide a citation for this particle range

Three papers measuring the particle size have been added ((Briggs et al. 2003; Dorfman et al. 1994; Fuller et al. 1997) to the manuscript, accordingly.

60: Add reference for Carlson, L.A. et al 2010. Plos Pathog. 6(11) and highlight cyro-EM on think cellular sections with viral budding.

This reference is concerning more the role of actin in HIV assembly; as we suppressed this idea from the review and narrowed it on HIV assembly and SRM, this reference was not added.

71-72: Why is the 7SL study important it seems to detract from the Gag vRNA focus of the review (simply, this reference is out of place)

We have removed the sentence and the reference there in.

76: Jouvenet et al 2008 Nature (ref 24) should be used here since this was the first example.  Remove reference 25 from this sentence.

This has been changed accordingly.

77-78: Remove reference 28 and add the earlier reference looking at HIV and ESCRT via TIRF: Jouvenet et al NCB, 2011, 13(4)

This has been changed accordingly.

81-85:  This is a fragmentary sentence and doesn’t make any logical sense.

This has been re-written.

110-130: Primary citations should come first and they reference the readers to a review on the subject at the end of the paper (section 2)

This has been corrected

123-125:

Figure 2: There is not inset for comparison between resolution improvements in a versus b. Can this improvement be quantified for the reader (simple linescan)?

The figure has been changed accordingly to the request of the reviewer. We introduced a c panel showing the improvement of “b” vs “a” resolution. We displaced the VLP to the “d” subsection of the figure and added the distribution of the VLP diameter obtained by PALM.

144-147: The authors must cite predecessor papers showing that the size and location of the fluorescent probe on the Gag polypeptide has been shown to perturb lattice assembly. Hubner et al. 2007, J Virol. 81(22), who was the first to create the internal MA-CA FP fusion for gag with GFP was careful to show that all the molecules of Gag in the cell cannot be tagged with the fluorescent protein. It would be important to make clear that doping with untagged Gag is critical to not perturbing the HIV assembly process. Are the images provided in reference 45 and in this manuscript performed in the presence of untagged Gag at suitable rations (>1:3 tagged to untagged)? This is also highlighted in papers from Jouvenet et al. 20xx.

This has been updated and highlighted in the new manuscript in section 2 (see lanes 170-177).

155-157: This sentence is not clear and doesn’t capture ref 50 conclusions accurately.

We have re-written this part of the manuscript in a more detailed way.

157-159: distinguish what present is relative to (e.g. is present at assembly sites?)

This has been removed from the text.

165: Reference VerPlank, L et al. PNAS 2001 for discovery of Tsg101 as p6 interactor.

The text has been changed and the use of this reference is no longer justified.

167-168: This sentence is very vague, what are the important details for the reader?

Indeed this sentence is not important for the reader. Therefore we removed it.

170: Remove the word “while” and replace with “as well”

This has been done.

160-171: The technologies used here must at least describe their performances. Why iPALM, it seems this reference uses 3D superresolution to directly measure the ESCRT positions, but it seems that Presher et al and Bleck et al used 2-dimensional imaging to infer the 3D position of ESCRTs. This is not clear.

This part of the manuscript has been rewritten. The first part deals with the interferometric PALM imaging and the benefit of a 3D super-resolution technique to measure ESCRT position and then the second part details the 2 colour, 2D SRM. We hope this makes this section clearer. 

175: This sentence doesn’t explain how this can open up possibilities. Of what? Very vague. This is not the first example of correlative iPALM and SEM (cited earlier in reference 52 line 162), what about this study improved possibilities?

This sentence has been changed for the following one: “All these studies show that SRM can now help in deciphering the respective positions of the different molecules involved in particle assembly in cells. Moreover, correlative iPALM and SEM studies by Van Engelenburg et al [67] and Pedersen et al [69] highlights the possibilities of SRM to shed light on a more resolved HIV-1 particle structure.” (see lanes 228-231).

177-181: No citations are used here to “bar a few”. This paragraph could be turned much more positive and highlight the strengths of the previous studies mentioned, but them transition to the extension to the temporal domains with superresolution imaging.

We have added citations to illustrate the “bare a few”. Moreover, we have rewritten this paragraph and tried to turn it more positively.

217: The term “aspects” is very vague. Be more specific: what aspects?

The term aspect has been removed and the sentence has been changed to: “Concerning Env incorporation into HIV-1 virions during particle assembly, it was recently studied by Buttler et al., [76] . Thanks to iPALM, it was shown that Env was incorporated in preformed Gag lattices to the neck of the assembling virions, as determined by the Env angular distribution on the surface of cell-associated virus.”  (see lanes 284-287).

223: This sentence makes it seem like these studies are not informative. Remove the words “could now” and “possible”.

We thank the reviewer for this remark, these words have been removed.

225: Define STED-FCS and perhaps highlight the strengths of this powerful technique (compare with SPT methods?).

STED-FCS has been defined the following way: “This SRM technique uses the fluorescence fluctuations of several molecules to measure their motion in each STED voxels of the line scanned (Honnigmann et al., 2014).” (lanes 292-293).

We think it is out of the scope of this review and it might be too technical to make a comparative study of the benefits and the drawback of sSTED-FCS and spt-PALM technique, thus we decided not to detail more.

226: Small is a very non-quantitative word. Word choice.

The word “small” has been changed to “low”

227: Use the word: “doubled”, “twice-increased” suggests that there are two doublings in steps.

We thank the reviewer for this remark. This has been changed in the text.

227-228: The comparison of diffusion coefficiencet between Chojnacki et al. and Buttler et al. are not correct. STED-FCS measurements of Env on the cell surface (Chojnacki et al) are in good agreement with those of Env diffusion via SPT (Buttler et al.). The mobile fraction on cells was determined via spt whereas the diffusion on released particles via STED-FCS was shown to be 100-fold lower, but only in comparison to Env diffusion on the cell surface (prior to lattice incorporation).

We thank the reviewer for his/her remark, we modified the manuscript accordingly and change the sentence to: “Interestingly, the Env diffusion coefficient found in Chojnacki et al. in HIV-1 particles [78] is a hundred times smaller than the one observed at the plasma membrane of virus producing cells [76,78], leaving an open window regarding the exact process and dynamics of Env incorporation during HIV-1 assembly.(see lanes 295-298). Because the mobile fraction is an empirical parameter depending on , time sampling of the technique used to monitor mobility, we prefer not discuss it and we decided to leave the proposal that there is still an open window to understand this 100x decrease of the Env diffusion.

Reviewer 3 Report

See attached

Author Response

Answers to Reviewer #3:

 The authors present a thorough review of recent work using single-molecule localization microscopy and other super-resolution modalities to study HIV assembly. My main critique is that in many cases, the review merely summarizes how individual studies were done, but it stops short of describing the results obtained and, perhaps more importantly, their impact on the field. Thus, it is not clear what key new knowledge was produced by these studies. Addressing the following comments should help alleviate this critique and after a throughout proofreading to fix several typos, this review should be publishable.

Specific Comments:

1. Line 65:

The Cys rich tag does not by itself enable visualization by fluorescence microscopy. How is the Cys tag labeled in these experiments?

We have added the details to be more accurate. The fluorescent tag was the Flash/ReAsh system. The sentence has been modified to: “In the past decade, the introduction of a small Cys rich tag into the viral Gag protein enabled the dynamic fluorescent imaging of Gag, using membrane-permeable biarsenical compounds FlAsH and ReAsH, in model cell lines and in macrophages [25,26].” (lanes 73-75).

2. Line 77:

o What is gRNA? Please define.

This has been defined in the text and in Figure 1, and we tried to keep it consistent throughout the review.

3. Line 85:

o There terms multifocus TIRF is an oxymoron. The Lamb group used a microscopy capable of “synchronous switching between TIRF and WF-microscopy on a frame by frame basis.” Please clarify in the text that it is either TIRF or multifocus WF microscopy, but both cannot be used at the same time.

We agree with the reviewer and thank him for pointing out this error. We have changed the sentence in the new manuscript (lane 82).

4. Line 99:

o Genomic RNA (vRNA)? Should it be abbreviated as gRNA?

Genomic RNA has been abbreviated when needed to gRNA throughout the manuscript. We thank the reviewer for noticing the inconsistency of the abbreviation in the overall text.

5. Line 104: “…the cortical actin network certainly plays a role in stabilizing the Gag assembly platform…”

“Certainly” seems too strong of an assertion, especially because the reader is not given enough context to come to that conclusion given what has been presented.

Because this review is not addressing or discussing the role of actin in HIV assembly, we removed the sentence from the main text, and choose to narrow the review only on HIV assembly and SRM techniques.

6. Line 114: “…can now be used to image cellular structures in three dimensions, with multiple colors, in living systems with nanometer-scale resolution…”

SRM methods can achieve that, but achieving all of these aspect at the same time is extremely challenging. Please qualify this statement.

We agree with the reviewer and therefore have changed the sentence in the following way: “Since their conception, super-resolution imaging methods have continually evolved and one could expect that they will allow imaging cellular structures in three dimensions, with multiple colours, in living systems at the nanometer scale resolution, resolution, even though it remains very challenging.” (lanes 124-126).

7. Line 128:

o the use of the term “correctly” is ambiguous. Do the authors mean accurately, precisely, or both?

This word has been changed to “precisely”.

8. Line 145: “The results show that Gag tagged with tdEOS forms unusually large clusters as compared to Gag-mEOS2, which in turn forms clusters well within the acceptable range for HIV-1 assembly sites.”

o Please interpret this statement for the reader.

Are the large clusters of tdEos and artifact of labeling?

What does this mean in terms of the biological mechanism?

We have rewritten this part of the manuscript. Indeed the comparison of td-EOS to m-EOS2 in the sentence clearly suggests that the large clusters are a consequence of td-EOS. We have added a sentence to explain the underlying possible biological mechanism:

Because the introduction of a protein tag into Gag could play a role, Gunzenhäuser et al. compared two photo-activable tags, mEOS2 and tdEOS, with respect to Gag assembly [51]. The results show that Gag tagged with tdEOS forms unusually large clusters as compared to Gag-mEOS2, which in turn forms clusters well within the range for HIV-1 assembly sites. Their data suggests that the addition of protein tags may very well change the nature of Gag assembly. The larger tandem dimeric tdEOS possibly disrupts the regular hexameric Gag lattice structure in the VLPs, changing Gag organization. “ (see lanes 163-169).

9. Line 150: “Using 3D SIM imaging, a recent study shows the presence of interacting genomic RNA in the cytosol and its spatial colocalization throughout the cell.”

o What is the genomic RNA interacting with? Please clarify.

We have deleted this sentence, and the corresponding reference, from the manuscript as this reference was not focus on HIV assembly and SRM per se but more on detecting the presence of the dimeric viral RNA genome into the cell cytosol, which is not the point of our review.

10. Line 153: “Dual color SRM and dSTORM have illustrated the importance of the CT domain of Env in HIV-1 Gag assembly.”

o What type of SRM?

We changed “SRM” to the more accurate “PALM/dSTORM” in the sentence:

“Dual color PALM/dSTORM and other dSTORM studies have illustrated the importance of the CT domain of Env in HIV-1 Gag assembly [63,64].” (lane 202-203).

11. Line 162: “They showed the initial scaffolding of ESCRT subunits CHMP2A, CHMP4B and TSG101 within the viral bud, the dynamics of which change dramatically on particle release.”

o Please interpret this statement for the reader.

How do the dynamics change?

Why is it dramatic? (I would suggest avoiding such emotional terms)

What does this mean for the biological mechanism(s)?

We have changed this part to better reflect the study cited. We have avoided the usage of hyperbolic terms and rewritten this part to include details for the mechanism. It now reads as follow:

“They showed the initial scaffolding of ESCRT subunits CHMP2A, CHMP4B and TSG101 within the viral bud followed by levels of CHMP2A decreasing significantly relative to Tsg101 and CHMP4B upon virus abscission. Apart from shedding light on the spatial distribution of ESCRT subunits within the particle, thanks to iPALM/correlative SEM, the differential incorporation of CHMP2A points to a distinct dynamic among the ESCRT subunits preceding viral abscission.” (lanes 215-217).

12. Line 177:

o This is the concluding paragraph of section 2. It would be good to summarize here what key new information has been learned by applying SRM to study HIV assembly. What question remain unanswered?

o The way this review is structured section 2 covers SRM work in fixed cells. However, some of the work is done in live cells. Would it make more sense to move these into section 3?

We tried to summarize the added value of SRM studies in visualizing HIV-1 assembly in cells. (lines 228-231) :

All these studies show that SRM can now help in deciphering the respective positions of the different molecules involved in viral particle assembly in cells. Moreover, correlative iPALM and SEM studies by Van Engelenburg et al [67] and Pedersen et al [69] highlights the possibilities of SRM to shed light on a more resolved HIV-1 particle structure. “

To emphasize the questions remaining to be solved, we add the following sentence:

“ In the last five years, many efforts have now been put into deciphering the real-time molecular events occurring in living cells during the HIV-1 assembly and budding process. “

Finally, except one study discussed in this section (ref 64), all the others were made on fixed cells. Therefore, we kept the same structure for section 2 and we dedicated the section 3 to real time single molecule assembly of HIV-1.

13. Line 189: “Recently, studies by Floderer et al. [34,45] in T cells and Yang et al. [58] in adherent cells used sptPALM to track and analyse Gag clusters during assembly and deciphered the role of the viral genomic RNA and the NC domain of HIV-1 Gag in this process.”

o Please explain what was deciphered?

o What does this mean for the biological mechanism(s)?

The sentence has been rewritten the following way (lanes 246-249):

“Recently, studies by Floderer et al. [38,48] in T cells and Yang et al. [72] in adherent cells used sptPALM to track Gag molecules and to confirm the important role of the gRNA and the NC domain of Gag in the generation of Gag clusters at the plasma membrane of the host cell for efficient HIV-1 assembly.”

Detailed information regarding the role of NC-RNA interaction in the initiation of assembly has been introduced from lanes 252 to 258.

14. Line 191: “By monitoring the trajectories of single HIV-1 Gag molecules in the vicinity of assembly sites both studies have shown the existence of directed motions towards these assembly sites.”

o What is the reason for directed motion?

At the moment there is no clear reason for this directed motion except that in the vicinity of the assembly site (less than 200 nm), there is an attraction of the surrounding Gag molecules. The strength of this attraction is measured in Floderer et al., We cannot state if this attraction comes from Gag multimerisation on the RNA, Gag-Gag interactions, membrane curvature ect… This is still an open question.

One sentence has been added: “It is currently not clear or elucidated what are these Gag directed motions towards the assembly site; it could reflect Gag multimerization on the viral RNA, Gag-Gag interactions, Gag-host cell factors interactions or else”. (lanes 254-256).

o Early in the review, the authors referred to the involvement of the cytoskeleton. Is there conclusive evidence for such involvement that could be included here?

Unfortunately, no. We have no evidence that actin could play a role in the attraction (directionality) of the assembly site surrounding Gag proteins.

15. Line 205:

o The reference should be to Figure 3e.

This has been updated. (line 269)

16. Line 205: “Interestingly, our results strongly suggested that any available cellular RNAs could lead to a complete assembly event with the same attraction energy than the vRNA, i.e. an energy of 3 to 4 kT (2-3 kcal/mol) on average.”

o So, full assembly is not specific for viral RNA???

This single molecule dynamics analysis showed that, as soon as an assembly seed has occurred (few Gag at the plasma membrane with an RNA molecule or two), the attraction energy senses by each Gag in the vicinity of the assembly site was, on average, equivalent either the cells were expressing Gag and the gRNA or Gag only. The “specificity” is not due to a change in an apparent binding constant related to this energy but to a temporal coordination of Gag attraction energy and density increase that is not seen when the encapsidable gRNA is absent.

17. Line 231:

o This is the concluding paragraph of section 3. It would be good to summarize here what key new information has been learned by applying live-cell SRM to study HIV assembly. What question remain unanswered?

 We feel that the conclusion, lane 302-306, is concise and sufficient because the questions that remain to be answered are addressed all along the main text.

Round  2

Reviewer 2 Report

The authors have done a sufficient job at addressing the majority of concerns.

Reviewer 3 Report

The authors have satisfactorily addressed all my critiques.  The manuscript should now be published.